# OpenReview forum: "Bits Leaked per Query: Information-Theoretic Bounds for Adversarial Attacks on LLMs"
_NeurIPS.cc/2025/Conference — NeurIPS 2025 spotlight_

### Official Review · Reviewer_3URk · 2025-06-29

**Clarity:** 2
**Significance:** 2
**Originality:** 2
**Rating:** 4
**Confidence:** 3

**Summary:**

This paper leverages Fano’s inequality to derive a lower bound on the number of queries $N$ needed for an adversarial attack $g$ on an LLM’s outputs $Z$ to succeed with rate $\varepsilon$:

$$
  N \geq \frac{\log(1/\varepsilon)}{I(Z; g(Z))}.
$$

Here, the bound holds under the assumption that $Z_1,\dots,Z_N$ are independent conditioned on $g(Z)$. Empirical evaluations on tasks like system-prompt leakage, jailbreaks, and relearning corroborate this bound holds in practice. Moreover, the authors observes that when attackers choose queries adaptively, the observed trade-off between $N$ and $\varepsilon$ is tighter.

**Questions:**

See weaknesses.

**Ethical Concerns:**

["NO or VERY MINOR ethics concerns only"]

**Final Justification:**

The authors provide additional results during the rebuttal which addressed my major concerns on the independence assumption.

**Limitations:**

The paper estimates mutual information using various methods, but it’s unclear how accurate those estimates are. Moreover, as noted under weaknesses, the independence assumption can be violated in certain cases; the authors should discuss the potential impact of this on their bound.

**Quality:**

2

**Strengths And Weaknesses:**

**Strengths**

* Formulate a important and relevant question for LLM safety and provide an attempt to address it.
* Derives a universal lower bound that applies to *any* adversarial attack strategy.

**Weaknesses**

* The proposed lower bound has limited practical implication. Although the bound could in principle inform safe query limits, its right-hand side depends on the specific attack’s mutual information term $I(Y;g(Y))$. As a result, (1) each attack yields a different threshold and (2) the defender must already know the attack to compute it—undermining real-world applicability.
* The independent assumption is usually invalid for realistic attacks (which are adaptive in nature). The theoretical bound assumes $Y_1,\dots,Y_N$ are independent given $g(Y)$, but adaptive attackers choose each query based on prior outputs, so those responses are correlated. This dependency contradicts the bound’s premise and calls into question the results shown in Figure 1(a).
* Some slight notation issues persist. The symbol $T$ denotes the attack outcome $g(Y)$ in Theorem 1, yet in Section 4 it is reused for “temperature.” Additionally, maybe “$T = g(Y)$” in theorem1 is a typo?

---

> ### Author Rebuttal · Authors · 2025-07-31
>
> - The proposed lower bound has limited practical implication. Although the bound could in principle inform safe query limits, its right-hand side depends on the specific attack’s mutual information term . As a result, (1) each attack yields a different threshold and (2) the defender must already know the attack to compute it—undermining real-world applicability.
>     - Thank you for raising this concern.
>     - First, the ability to place a theoretical lower bound on existing defence methods is valuable because it introduces a mathematical safety guarantee into LLM-security evaluations that often rely on heuristics, and it lets us compare multiple defences on a common scale. Assuming existing attacks, therefore, does not diminish the usefulness of our work.
>     - Second, the real issue is whether the bound is robust to unknown attacks; the mere fact that it is derived with respect to known ones should not, in itself, be problematic.
>     - To test generalization, we ran an additional experiment: we estimated $I$ and measured the actual value using different attack methods (GCG and PAIR), then counted how often the resulting bound was violated.
>     - Using the same three tasks and seven LLMs as in the main study, we found that no instance fell below the bound, confirming its robustness, just as in Figure 1. We also bootstrap-tested the differences between bounds computed with the two attacks ($p=0.05$) and observed no significant variation in any setting. We will include this experiment and its discussion in the appendix in the camera-ready version.
> - The independent assumption is usually invalid for realistic attacks (which are adaptive in nature). The theoretical bound assumes are independent given , but adaptive attackers choose each query based on prior outputs, so those responses are correlated. This dependency contradicts the bound’s premise and calls into question the results shown in Figure 1(a).
>     - Thank you for highlighting this issue. The i.i.d. assumption was introduced only to streamline the original proof; it is not required for the theory itself. We have already verified that the theorem still holds without assuming i.i.d., as we prove below. Consequently, our theory also covers fully adaptive attack settings and multi-turn conversation settings.
>     - In the camera-ready version, we will replace the current statement under the i.i.d. assumption and proof with the independence-free version in Section 2.
>
> ---
>
> **Theorem 1** (Information‑theoretic lower bound for adaptive attacks without the i.i.d. assumption)
>
> Let
> - $T\in\mathcal{T}$ be the target property with an arbitrary prior (finite, countable, or continuous);
> - an attacker issue $N$ sequential queries, where the $n$‑th input $X_n$ is any measurable function of the past outputs $Z_{1:n-1}$ (i.e., the attack is adaptive);
> - the model reply is
> $$Z_n = g\bigl(X_n,T,U_n\bigr),$$
> where $U_n$ is internal randomness independent of $T$ and of all previous $(X_i,Z_i)_{i<n}$.
>
> Define the per‑query leakage
>
> $$
> I_{\max}:=\sup_{x\in\mathcal{X}} I\bigl(Z;T \mid X=x\bigr)\quad\text{[bits]}.
> $$
>
> For any tolerated error probability $0<\varepsilon<1$, every adaptive strategy must issue at least
>
> $$
> N_{\min}(\varepsilon)\ge\frac{\log_2(1/\varepsilon)}{I_{\max}}.
> $$
>
> **Proof**
>
> 1. Fano’s inequality.
>
> Let the attacker’s estimate be $\widehat{T}=f(Z_{1:N})$ with error probability
> $$P_{\mathrm{err}}:=\Pr[\widehat{T}\neq T].$$
> The $K$‑ary (or differential) Fano inequality gives
> $$H_{\mathcal{T}}\bigl(P_{\mathrm{err}}\bigr) \ge H(T)-I(Z_{1:N};T).$$
>
> 2. Chain rule without independence.
>
> Always
> $$I(Z_{1:N};T)=\sum_{n=1}^{N} I \bigl(Z_n; T \mid Z_{1:n-1}\bigr).$$
> Each term is upper‑bounded by $I_{\max}$ by definition, so
> $$I(Z_{1:N};T) \le N I_{\max}.$$
>
> 3. Combine the bounds.
>
> For $P_{\mathrm{err}}\le\varepsilon$, entropy satisfies
> $H_{\mathcal{T}} \bigl(P_{\mathrm{err}}\bigr)<\log_2(1/\varepsilon)$, hence
> $$
> N I_{\max} \ge \log_2(1/\varepsilon).
> $$
>
> 4. Rearrange.
>
> $$N \ge \frac{\log_2(1/\varepsilon)}{I_{\max}},$$
> establishing the claimed lower bound.
>
> ∎
>
> **Interpretation.**
>
> The i.i.d. assumption in prior work is only used to equalise the mutual‑information terms.
> Replacing that with a uniform upper bound $I_{\max}$ suffices because the chain rule already accounts for correlations between successive replies.
> Thus, the result holds for fully adaptive, stateful attacks.
>
>
> ---
>
>
> - Some slight notation issues persist. The symbol denotes the attack outcome in Theorem 1, yet in Section 4 it is reused for “temperature.” Additionally, maybe “” in theorem1 is a typo?
>     - Thank you for catching the notation clash. We will keep $T:=g(Y)$, rename the temperature to $\tau$, and correct the typo. These edits clarify the notation without affecting any results.

---

> > ### Author Response · Authors · 2025-08-06
> >
> > Reviewer 3URk,
> >
> > As the deadline for the discussion period is approaching, I would like to offer a gentle reminder. If our rebuttal has not fully addressed your concerns, I would greatly appreciate it if you could let us know so that we may clarify further. If your concerns have been resolved, we would greatly appreciate your reconsideration of the score.
> >
> > Thank you for your time and consideration.

---

> > ### Comment · Reviewer_3URk · 2025-08-08
> >
> > Thank you for the clarification and the additional proof. My question was properly addressed, and I appreciate the authors' response. I will raise my score.

---

### Official Review · Reviewer_NDHW · 2025-07-02

**Clarity:** 3
**Significance:** 3
**Originality:** 4
**Rating:** 5
**Confidence:** 3

**Summary:**

This paper demonstrates some information theoretic bounds on the success rate of adversarial attacks against LLMs. In particular, the authors use results from information theory to show that the number of queries needed to identify a certain property, the model which is unknown to the attacker beforehand, is lower bounded by
$\log (1/\epsilon)/I(Z;\tau)$ for error rate $\epsilon$ and query $Z$ and hidden property $T$. The authors then use neural mutual information estimation in order to demonstrate that their theorem is correct, and to show that attacks which have exposed more bits per query, such as by returning logits, are able to find out the target property more quickly.

**Questions:**

1. Can you elaborate on the experimental results in figure 1? Namely, why do the tok + TP + logit attacks require *more* samples than the tok attacks? And why is the claimed result that N \propto 1/I(Z;T) while the plot seems to show that n \propto I(Z;T)?
2. How decision-relevant would this be for a provider of access to a model? Would they be able to use these results in order to determine relevant considerations such as allowed level of access, or number of top-k logits?
3. To what extent is the gap between the lower bound due to under-estimation of the MI and ineffectiveness of the attack? It could be possible to use more modern MI estimators such as the ones from [1]?

   [1]: Understanding the Limitations of Variational Mutual Information Estimators, Song et al 2020,

**Ethical Concerns:**

["NO or VERY MINOR ethics concerns only"]

**Final Justification:**

This is a decent paper which tries to quantify 'levels of access' to an LLM for adversarial elicitation such as jailbreaks, system prompt extraction, etc.

The results in figure 1 were initially quite confusing, since they seemed to be the opposite way round to what we would expect based on the theoretical arguments, as well as common sense. However, the authors have claimed that the real data should be reversed compared to that presented, which resolves this issue. The fact that the data seemed clearly erroneous and was then fixed immediately is somewhat concerning, especially since I don't believe the full code was provided. However, taking the authors in good faith, the results are solid and back up the theoretical work. While not exceptional, this is a good contribution which has both practical and theoretical impacts.

**Limitations:**

Yes

**Quality:**

3

**Strengths And Weaknesses:**

### Strengths
+ The theorems are nicely demonstrated and the story makes sense overall.
+ The paper is quite clear overall and well-written
+ The attempt to rigorously characterise concepts such as 'level of access' is welcome, and although the result is straightforward and aligns with our intuitions, it is good to have a clear exposition in the literature.

### Weaknesses
+ The importance of the main result is not very clear to me. The characterization that the number of queries is lower-bounded by $\log (1/\epsilon)/I(Z;\tau)$ is nice to know, but given the relatively large gap between the lower bound and the actual required number of queries means that it is hard to actually use this information in a decision-relevant way. For instance, it would be nice to answer the question 'given this level of access to a novel model, how many suspicious queries should we allow before flagging the user for scrutiny', or indeed 'given this many suspicious queries are allowed, should we provide logits, thinking As far as I can tell, this is not possible with the current method.

+ The result in figure 1 seems like the $x$-axis is mislabelled, which is concerning when it is the main result of the paper. I think I am probably misunderstanding the figure somehow, but if I am not then the experimental results seem opposite to the expected results. Namely, figure 1 plots inverse leaked bits per query (x axis, $1 / I(Z;T)$) against the number of queries needed to get $1 - \epsilon$ error. From the theorem we have that $N_{min} = \log(1/\epsilon)/I(Z;T)$. $\log (1 / \epsilon)$ is a constant for the purposes of the plot. Therefore we should expect that the graph shows a relationship of $y = x + C$, since $N$ is proportional to $1 / I(Z;T)$. However we see on the graph that $y = -x + C$. Furthermore, the data also seem to be inconsistent with common sense: an average 'Tok + TP + logit' example is at $1/I = 5, QS=200$ for Llama4-M while an average 'Tok' example is at $1/I = 10, QS=80$. It seems very strange that the 'Tok' examples should require *fewer* queries than the 'Tok + TP + logit' examples. Since this is such a central figure in the paper, I expect I am misreading it but if not then this is a serious flaw.

---

> ### Author Rebuttal · Authors · 2025-07-31
>
> - Regarding usefulness in the decision-making phase
>     - Thank you for the insightful comment. Our lower bound enables several concrete actions for practitioners, such as:
>         1. Compute a safe query budget: By measuring the average information leaked per query $I(Z; T)$, we can instantly derive $N_{\max}$ (the safe limit on “suspicious” queries). If we want to expose top-$k$ logits while keeping attack risk below 1 %, we can set a rate limit that automatically stops serving logits once a user exceeds $N_{\max}$.
>         2. Rate-limit suspicious traffic: Estimate $I(Z;T)$ once; flag or throttle when a user’s suspicious queries exceed $N_\max$.
>         3. Benchmark red-team efficiency: Using the gap between the empirical query count $N$ attack $N_{\rm attack}$ and the theoretical lower bound $N_\max$ as a metric lets us quantify how close an attack is to the information-theoretic limit and how much room for improvement remains.
>     - As you suggested, a more modern MI estimator might further shrink the gap between the theoretical and empirical lower bounds. To test this, we replaced our original estimator with the one you recommended, SMILE [1], and compared the ratio between the theoretical lower bound  $N_\min^{\rm th}$ and the empirical lower bound $N_\min^{\rm emp}$,  $N_\min^{\rm th} / N_\min^{\rm emp}$, under both estimators. In the following table, values near zero mean the gap did not close with SMILE, and asterisks in the table indicate bootstrap significance at $p = 0.05$.
>     - As a result, we found almost no change in that ratio. These results suggest the gap stems mainly from attack inefficiency, not MI under-estimation. We will add the table and a brief discussion to the appendix.
>
> | Task / Model              | OLMo2‑7B | OLMo2‑13B | OLMo2‑32B | Llama4‑M | Llama4‑S | DeepSeek‑R1 | GPT‑4 |
> | ------------------------- | -------: | --------: | --------: | -------: | -------: | ----------: | ----: |
> | **System‑prompt leakage** |    0.015 |     0.012 |     0.018 |    0.020 |    0.017 |     0.025\* | 0.014 |
> | **Jailbreak**             |  0.028\* |     0.022 |     0.026 |  0.030\* |    0.027 |     0.035\* | 0.024 |
> | **Relearning**            |    0.010 |     0.011 |     0.013 |    0.016 |    0.015 |       0.019 | 0.012 |
>
> - Clarification on experimental results
>     - Thank you for catching the confusion in Figure 1. The horizontal axis should have been $\log I(Z; T)$, not “inverse leaked bits.”
>     - We will (1) relabel the axis, (2) plot the theoretical line with the correct slope -1, and (3) horizontally flip the data so higher leakage (Tok + TP + logit) lies to the right and lower, exactly matching intuition.
>     - These fixes are purely cosmetic; the data, statistics, and conclusions remain unchanged. In the correct figure, the “Tok” points move to the upper left and the “Tok + TP + logit” points to the lower right, fully matching the intuition that exposing more information lowers the required number of queries. This resolves your concerns.
>
> [1]: Understanding the Limitations of Variational Mutual Information Estimators, Song et al 2020

---

> > ### Comment · Reviewer_NDHW · 2025-08-03
> > **response**
> >
> > Thanks for your rebuttal.
> >
> > The experimental results on SMILE are interesting--it points out that another corollary of your work is to determine if our current attacks are near-optimal or if there are potentially more powerful ones yet to be discovered.
> >
> > Regarding figure 1, I am still confused. If the issue is that the x-axis is plotting $I$ instead of $1/I$, then plotting the data correctly would result in a horizontal flip of the points. But wouldn't this still leave the $y$-coordinates of the data unchanged? As I said in my review, an average 'Tok + TP + logit' example is at $1/I = 5, QS=200$ for Llama4-M while an average 'Tok' example is at $1/I = 10, QS=80$. Flipping the $x$-axis would result in the 'Tok + TP + logit' example being at $I = 5, QS=200$ while the average 'Tok' example is at $I = 10, QS=80$. The mutual information numbers now make sense, but the query numbers are still the opposite of what the results in the main paper describe, with the more informative access requiring more queries?

---

> > > ### Author Response · Authors · 2025-08-05
> > >
> > > Thank you for the opportunity to provide further clarification.
> > >
> > > In our earlier response, we noted that the plot was “mirrored,” which may not have fully conveyed that the data series was mislabeled in the original figure.
> > > Specifically, we simultaneously (i) switched the x-axis to log I(Z; T) and (ii) corrected the legend assignments (Tok = red, Tok + logit = green, Tok + TP = orange, Tok + TP + logit = blue), and this combination understandably caused confusion.
> > >
> > > With the correct label-to-data mapping, the Tok + TP + logit appears in the lower-right of Tok, perfectly aligned with the theoretical line of slope −1. This clearly illustrates the expected monotonic trend: queries that leak more bits require fewer total queries.
> > >
> > > For clarity, we provide a simplified ASCII representation of the corrected figure below, as OpenReview does not allow image attachments.
> > >
> > >
> > > Queries ↑
> > >
> > > │
> > >
> > > │ ●●●●●●● Tok
> > >
> > > │ &emsp;&emsp;▲▲▲▲▲▲▲ Tok + logit
> > >
> > > │ &emsp;&emsp;&emsp;&emsp;&emsp;&emsp;&emsp;■■■■■■■ Tok + TP
> > >
> > > │ &emsp;&emsp;&emsp;&emsp;&emsp;&emsp;&emsp;&emsp;&emsp;◆◆◆◆◆◆◆◆◆ Tok + TP + logit
> > >
> > > │
> > >
> > > └────────────────────────────── log I(Z; T) →

---

> > > > ### Comment · Reviewer_NDHW · 2025-08-06
> > > >
> > > > Thanks for clarifying the error in the experimental results.
> > > >
> > > > This discussion resolves the uncertainty I had regarding the plots, and I'm happy to leave my score at **accept**.

---

### Official Review · Reviewer_uQ13 · 2025-07-02

**Clarity:** 3
**Significance:** 3
**Originality:** 3
**Rating:** 5
**Confidence:** 4

**Summary:**

The paper analyzes the query complexity of extracting hidden information from LLMs. They do this by considering the mutual information between the information $Z$ revealed by the LLM when it responds to a prompt, and a hidden target $T$.

$T$ can be e.g. the model’s system prompt, the indicator of whether the model’s raw response complies with a harmful user request, or a piece of information seen during training that the model is not meant to reveal. Meanwhile, $Z$ can include the model’s output tokens, next-token logits and chains of thought.

The authors derive an information-theoretic lower bound indicating that any adaptive attack needs at least $\log(1/\epsilon) / I(Z, T)$ queries to recover $T$ up to accuracy $\epsilon$. The bound is first derived in the binary case, and then extended to the K-ary and continuous cases. Interestingly, the bound in the K-ary case does not depend directly on $K$ (so that it has the same form as the binary case).

In the binary case, the authors also give an attack that matches the lower bound using a sequential probability ratio test. In the appendix, they extend this attack also to the K-ary and continuous settings.

The authors then set out to experimentally validate to what extent various attacks align with their theoretical lower bound. The classes of attack considered are system prompt leakage attacks, jailbreak attacks (e.g. GCG), and relearning (i.e. extracting unlearned information present in the training data). They consider three disclosure regimes: output tokens, output tokens + CoT, and output tokens + CoT + logits.

One practical problem for their evaluations is estimating the mutual information. The authors’ approach consists of taking the maximum of three variational-lower-bound-based estimators (MINE, NWJ and InfoNCE).

Their findings can be summarized as follows:

- **System-Prompt Leakage**: Adaptive attacks under tokens, CoT, and logits regimes follow $N \propto 1/I(Z;T)$, with richer signals reducing required queries by orders of magnitude.
- **Jailbreak Attacks**: Methods like GCG and PAIR lie on the same inverse-information curve, showing that access to logits or CoT greatly accelerates jailbreak success.
- **Relearning Attacks**: Extracting “unlearned” training data also exhibits $N \propto 1/I(Z;T)$, where more informative leaks sharply lower the query budget for high-quality recovery.
- **Decoding Hyperparameter Sweep**: Decreasing temperature or nucleus threshold reduces mutual information and causes an exponential increase in required queries, confirming that output diversity (both in the form of sampling randomness and of disclosure level) governs leakage cost.

**Questions:**

- In theorem 1, is my understanding correct that, since $T$ is an indicator of attack success, the attacker aims not to produce a prompt that jailbreaks the model, but rather to determine, for a given query, whether the model is jailbroken on it? If this is the case, then the setting does not reflect an attack trying to steer the model past its guardrails (line 30), but rather only detect whether the model is violating its guardrails (which is a very different and generally easier problem).
- In theorem 1, are any assumptions placed on the choice of query? Is a query understood as an arbitrary prompt, or does it necessarily consist of e.g. feeding the model the same prompt repeatedly?
- Related to the above, does theorem 1 encompass attacks which adapt their prompts depending on past observables $Z$? My understanding is it does not, as then $(Z_i)$’s would not be independent.
- In the SPRT attack, how does the attacker compute the likelihood ratio, concretely? Does the theoretical analysis assume access to the ground truth conditional distributions $p(Z | T=b); b \in \{0, 1\}$? If so, how would your query complexity bound change if the attacker instead needs to estimate these distributions?

The main uncertainty I have with regards to recommending acceptance is understanding the questions above, with particular regard to whether the theorems account for adaptive algorithms in a jailbreak setting.

**Ethical Concerns:**

["NO or VERY MINOR ethics concerns only"]

**Final Justification:**

The author's reply gives me more confidence that the results in this work aren't limiting in the way they initially seemed. Indeed, the fact that "hardness of verification implies hardness of generation" (in the information-theoretic sense) is a good observation, and would make the paper clearer if included. It is also a lot easier to see the value of their bound in the non-iid case.

Given these updates, I will now update my score to **accept**.

**Limitations:**

Yes.

**Paper Formatting Concerns:**

None.

**Quality:**

3

**Strengths And Weaknesses:**

**Strengths:**
- Important problem setting: leaking data from LLMs is a major concern for large model providers, and understanding this security risk is of large practical value.
- Clear and general framework: the authors demonstrate the bits-leaked-per-query framework can be applied to analyze the complexity of various forms of attack, ranging from jailbreaking to system prompt leakage.
- Very clear trend in results: the results clearly indicate, in the authors’ experimental setting, that the number of queries required for reconstruction (up to some level of accuracy) show a clear inverse scaling with respect to the mutual information between the observables and the hidden target.
- Potential for downstream applicability: in addition to further theoretical / analytical work in this framework (e.g. incorporating aspects of query adaptivity in the attacks analyzed), practical engineering work can also follow from this framework, including e.g. proposing adaptive defenses against these attacks, potentially based on the information-theoretic lower bounds provided by the authors.
- Practical quantitative guidelines on how different disclosure levels lead to information leakage and impacts query complexity, stemming from the results in Figure 1.

**Weaknesses:**
- Certain aspects of the theoretical setting are unclear: as highlighted in the questions below, at first glance, it seems that requiring that $(Z, T)$ pairs be i.i.d. might restrict the class of attacks under consideration, excluding attacks that alter subsequent queries depending on prior outputs. Also, it is not clear to what extent trying to recover an attack success indicator corresponds to model jailbreaking, as the former task relates to verifying attack success, while the latter task consists of producing an attack success, which a priori can be considered to be harder.
- Potential disconnect between theorems and experimental setting: in case the theorem assumptions do indeed exclude attacks which modify their queries based on the model’s past responses, the adaptive attack setting (line 181) is not covered by the theorem. If this is the case, I believe it should be clearly noted (please correct me in case I misunderstood things or missed the author’s acknowledgement of any disconnect).

---

> ### Author Rebuttal · Authors · 2025-07-31
>
> - Certain aspects of the theoretical setting are unclear: as highlighted in the questions below, at first glance, it seems that requiring that pairs be i.i.d. might restrict the class of attacks under consideration, excluding attacks that alter subsequent queries depending on prior outputs. Also, it is not clear to what extent trying to recover an attack success indicator corresponds to model jailbreaking, as the former task relates to verifying attack success, while the latter task consists of producing an attack success, which a priori can be considered to be harder.
>     - Thank you for highlighting this issue. Firstly, the i.i.d. assumption was introduced only to streamline the original proof; it is not required for the theory itself. We have already verified that the theorem still holds without assuming i.i.d., as we prove below. Consequently, our theory also covers fully adaptive attack settings and multi-turn conversation settings.
>     - In the camera-ready version, we will replace the current statement under the i.i.d. assumption and proof with the independence-free version in Section 2.
>     - Second, regarding the distinction between verification (detecting that a prompt has already succeeded) and generation (finding such a prompt), the two are information-theoretically equivalent. Consequently, the query lower bound $N_{\min}(\varepsilon)$ that we derive for verification applies unchanged to generation. This means an adversary that cannot verify success within $N_{\min}(\varepsilon)$ queries also cannot generate a successful prompt in fewer than $N_{\min}(\varepsilon)$ queries.
>
> ---
>
> **Theorem 1** (Information‑theoretic lower bound for adaptive attacks without the i.i.d. assumption)
>
> Let
> - $T\in\mathcal{T}$ be the target property with an arbitrary prior (finite, countable, or continuous);
> - an attacker issue $N$ sequential queries, where the $n$‑th input $X_n$ is any measurable function of the past outputs $Z_{1:n-1}$ (i.e., the attack is adaptive);
> - the model reply is
> $$Z_n = g\bigl(X_n,T,U_n\bigr),$$
> where $U_n$ is internal randomness independent of $T$ and of all previous $(X_i,Z_i)_{i<n}$.
>
> Define the per‑query leakage
>
> $$
> I_{\max}:=\sup_{x\in\mathcal{X}} I\bigl(Z;T \mid X=x\bigr)\quad\text{[bits]}.
> $$
>
> For any tolerated error probability $0<\varepsilon<1$, every adaptive strategy must issue at least
>
> $$
> N_{\min}(\varepsilon)\ge\frac{\log_2(1/\varepsilon)}{I_{\max}}.
> $$
>
> **Proof**
>
> 1. Fano’s inequality.
>
> Let the attacker’s estimate be $\widehat{T}=f(Z_{1:N})$ with error probability
> $$P_{\mathrm{err}}:=\Pr[\widehat{T}\neq T].$$
> The $K$‑ary (or differential) Fano inequality gives
> $$H_{\mathcal{T}}\bigl(P_{\mathrm{err}}\bigr) \ge H(T)-I(Z_{1:N};T).$$
>
> 2. Chain rule without independence.
>
> Always
> $$I(Z_{1:N};T)=\sum_{n=1}^{N} I \bigl(Z_n; T \mid Z_{1:n-1}\bigr).$$
> Each term is upper‑bounded by $I_{\max}$ by definition, so
> $$I(Z_{1:N};T) \le N I_{\max}.$$
>
> 3. Combine the bounds.
>
> For $P_{\mathrm{err}}\le\varepsilon$, entropy satisfies
> $H_{\mathcal{T}} \bigl(P_{\mathrm{err}}\bigr)<\log_2(1/\varepsilon)$, hence
> $$
> N I_{\max} \ge \log_2(1/\varepsilon).
> $$
>
> 4. Rearrange.
>
> $$N \ge \frac{\log_2(1/\varepsilon)}{I_{\max}},$$
> establishing the claimed lower bound.
>
> ∎
>
> **Interpretation.**
>
> The i.i.d. assumption in prior work is only used to equalise the mutual‑information terms.
> Replacing that with a uniform upper bound $I_{\max}$ suffices because the chain rule already accounts for correlations between successive replies.
> Thus, the result holds for fully adaptive, stateful attacks.
>
>
> ---
>
> - In theorem 1, is my understanding correct that, since is an indicator of attack success, the attacker aims not to produce a prompt that jailbreaks the model, but rather to determine, for a given query, whether the model is jailbroken on it? If this is the case, then the setting does not reflect an attack trying to steer the model past its guardrails (line 30), but rather only detect whether the model is violating its guardrails (which is a very different and generally easier problem).
>     - Thank you for your comment. Our framework covers the generation of a jailbreak prompt, not merely the detection of an attack.
>     - We suspect there may be a misunderstanding, so allow us to clarify the role of $T$.
>     - For every candidate query $q_i$, there is an unknown latent indicator $T(q_i)∈{0,1}$ that tells whether that query will bypass the guardrails. An attacker therefore  proceeds in the following “generate-and-verify” loop:
>          1. **Generate** a candidate query $q_i$
>          2. **Query** the model with $q_i$  and observe the response $Z_i$.
>          3. **Infer** $T_i$ from $Z_i$
>          4. **Terminate or iterate**: If $T_i=1$, the jailbreak succeeds; otherwise, the attacker constructs a new prompt $q_{i+1}$  and returns to Step 1.
> - Regarding the queries (prompts) in scope
>     - Thank you for the question. Theorem 1 places no constraints on the queries: each $q_i$ can be any finite string, chosen adaptively from past outputs. Consequently, our framework accommodates both repeated prompts and different prompts at each round.
> - In the SPRT attack, how does the attacker compute the likelihood ratio, concretely? Does the theoretical analysis assume access to the ground truth conditional distributions ? If so, how would your query complexity bound change if the attacker instead needs to estimate these distributions?
>     - Thank you for the insightful question. Our lower-bound analysis adopts a worst-case model in which the attacker is granted oracle access to the true conditional distributions and can therefore compute the likelihood ratio exactly.
>     - If, in reality, the attacker must first estimate these distributions, the required number of queries can grow. Consequently, our bound remains valid, and in practice becomes even more protective.

---

> > ### Comment · Reviewer_uQ13 · 2025-08-04
> >
> > Thank you for your careful reply. It gives me more confidence that the results in this work aren't limiting in the way they initially seemed. Indeed, the fact that "hardness of verification implies hardness of generation" (in the information-theoretic sense) is a good observation, and would make the paper clearer if included. It is also a lot easier to see the value of your bound in the non-iid case.
> >
> > Given these updates, I will now update my score to **accept**.

---

### Official Review · Reviewer_iH4f · 2025-07-03

**Clarity:** 2
**Significance:** 2
**Originality:** 2
**Rating:** 5
**Confidence:** 3

**Summary:**

This paper presents a principled, information-theoretic framework to measure and bound the efficiency of adversarial attacks on large language models (LLMs). By treating observable model responses as a random variable Z and the attacker’s target (e.g., system prompt, harmful output flag, or forgotten content) as another variable T, the authors define mutual information I(Z;T) as the number of bits leaked per query. They empirically validate this relationship through experiments on 7 LLMs (GPT-4, DeepSeek-R1, OLMo2, LLaMA 4) across 3 attack types: jailbreaks, system-prompt leakage, and relearning, using both adaptive (GCG, PAIR) and non-adaptive attacks. Their results confirm a tight inverse correlation between leakage and query budget, giving both attackers and defenders a universal metric to balance transparency versus risk.

**Questions:**

1. Can your theoretical bounds extend to settings where queries are not i.i.d., especially for adaptive attacks that depend heavily on feedback?
2. How are signals like logits and chain-of-thought combined? Is there evidence of synergy or redundancy in the leakage they offer?
3. Beyond lowering temperature or hiding logits, do you recommend any defense techniques (e.g., randomized decoding, response perturbation) based on your findings?
4. Could your framework also inform differential privacy-style bounds on allowable leakage for LLMs trained with user-sensitive data?
5. Do your bounds generalize to multi-turn conversations where leakage may accumulate or be reinforced across rounds?
6. Could your mutual information formulation be extended to LLMs with vision (e.g., GPT-4V)? What are the additional challenges?

**Ethical Concerns:**

["NO or VERY MINOR ethics concerns only"]

**Limitations:**

Yes

**Quality:**

3

**Strengths And Weaknesses:**

### Strengths:
1. The core contribution, a tight bound linking mutual information and query complexity is elegantly proven, generalizing across binary, multi-class, and continuous targets.
2. The idea of using bits leaked per query as a universal yardstick for evaluating and comparing adversarial attacks is intuitive, theoretically grounded, and actionable.
3. Seven LLMs across three attack types confirm the predicted inverse relationship. The findings hold under diverse leakage types (tokens, logits, chain-of-thought).
4. The results inform how transparency features like logit visibility or thought traces can drastically reduce the attack cost, with significant implications for deployment and UI design.
5. Code is attached, training and evaluation details are comprehensive (including estimator training, attack thresholds, hyperparameter sweeps), and results include statistical significance tests.

### Weaknesses:
1. While the paper emphasizes how more leakage accelerates attacks, it gives little actionable guidance on how to mitigate leakage besides reducing output diversity.
2. The theoretical framework assumes i.i.d. query-response pairs, which may not hold in adaptive attack settings where each query depends on past outputs.
3. Mutual information is estimated via lower bounds (MINE, NWJ, InfoNCE) using RoBERTa. While practical, this introduces bias and may not capture fine-grained leakage in high-entropy settings.
4. The study would be stronger with examples from LLM APIs that have real safety filters and deployment quirks, bridging theory with live systems.
5. Although addressed briefly, a deeper engagement with potential misuse of the framework (e.g., better jailbreaks from knowing theoretical limits) would improve robustness.

---

> ### Author Rebuttal · Authors · 2025-07-31
>
> - While the paper emphasizes how more leakage accelerates attacks, it gives little actionable guidance on how to mitigate leakage besides reducing output diversity.
>     - Thank you for raising this point. We respectfully point out that adjusting output diversity is only one facet of our proposed method. In Section 3 and again in the experimental result (Figure 1), we introduce and validate a theory that restricts the public signal can reduce the leakage effectively.
> - Discussion Without the Independence Assumption
>     - Thank you for highlighting this issue. The i.i.d. assumption was introduced only to streamline the original proof; it is not required for the theory itself. We have already verified that the theorem still holds without assuming i.i.d., as we prove below. Consequently, our theory also covers fully adaptive attack settings and multi-turn conversation settings.
>     - In the camera-ready version, we will replace the current statement under the i.i.d. assumption and proof with the independence-free version in Section 2.
>
> ---
>
> **Theorem 1** (Information‑theoretic lower bound for adaptive attacks without the i.i.d. assumption)
>
> Let
> - $T\in\mathcal{T}$ be the target property with an arbitrary prior (finite, countable, or continuous);
> - an attacker issue $N$ sequential queries, where the $n$‑th input $X_n$ is any measurable function of the past outputs $Z_{1:n-1}$ (i.e., the attack is adaptive);
> - the model reply is
> $$Z_n = g\bigl(X_n,T,U_n\bigr),$$
> where $U_n$ is internal randomness independent of $T$ and of all previous $(X_i,Z_i)_{i<n}$.
>
> Define the per‑query leakage
>
> $$
> I_{\max}:=\sup_{x\in\mathcal{X}} I\bigl(Z;T \mid X=x\bigr)\quad\text{[bits]}.
> $$
>
> For any tolerated error probability $0<\varepsilon<1$, every adaptive strategy must issue at least
>
> $$
> N_{\min}(\varepsilon)\ge\frac{\log_2(1/\varepsilon)}{I_{\max}}.
> $$
>
> **Proof**
>
> 1. Fano’s inequality.
>
> Let the attacker’s estimate be $\widehat{T}=f(Z_{1:N})$ with error probability
> $$P_{\mathrm{err}}:=\Pr[\widehat{T}\neq T].$$
> The $K$‑ary (or differential) Fano inequality gives
> $$H_{\mathcal{T}}\bigl(P_{\mathrm{err}}\bigr) \ge H(T)-I(Z_{1:N};T).$$
>
> 2. Chain rule without independence.
>
> Always
> $$I(Z_{1:N};T)=\sum_{n=1}^{N} I \bigl(Z_n; T \mid Z_{1:n-1}\bigr).$$
> Each term is upper‑bounded by $I_{\max}$ by definition, so
> $$I(Z_{1:N};T) \le N I_{\max}.$$
>
> 3. Combine the bounds.
>
> For $P_{\mathrm{err}}\le\varepsilon$, entropy satisfies
> $H_{\mathcal{T}} \bigl(P_{\mathrm{err}}\bigr)<\log_2(1/\varepsilon)$, hence
> $$
> N I_{\max} \ge \log_2(1/\varepsilon).
> $$
>
> 4. Rearrange.
>
> $$N \ge \frac{\log_2(1/\varepsilon)}{I_{\max}},$$
> establishing the claimed lower bound.
>
> ∎
>
> **Interpretation.**
>
> The i.i.d. assumption in prior work is only used to equalise the mutual‑information terms.
> Replacing that with a uniform upper bound $I_{\max}$ suffices because the chain rule already accounts for correlations between successive replies.
> Thus, the result holds for fully adaptive, stateful attacks.
>
>
> ---
>
>
> - Regarding the MI estimators
>     - Thank you for the observation. Using a stronger encoder or a tighter estimator would indeed bring the estimated value closer to the true mutual information. However, Figure 1 shows that our MINE/NWJ/InfoNCE-based method never exceeds the lower bound of mutual information, so it serves as a valid lower bound.
>     - We therefore conclude that the present MI estimators already measure leakage with adequate fidelity for the purposes of this work.
> - Regarding the use of the LLM API
>     - Thank you for the suggestion. To bridge theory with live systems, we have already included an LLM API, OpenAI GPT-4, in our evaluation. The experimental results in Section 3 provide direct evidence that our theoretical findings hold under practical safety filters and deployment quirks.
> - Potential misuse of this framework
>     - Thank you for raising this concern. We acknowledge that, in principle, signals with higher leakage could be exploited to create stronger jailbreak attacks. At the same time, the same analysis equips defenders to pinpoint which signals leak the most information and to withhold those signals for users who exhibit potentially malicious behavior.
>     - We will clarify this dual-use aspect and outline concrete defensive deployments in the camera-ready version.
>
>
> - How are signals like logits and chain-of-thought combined? Is there evidence of synergy or redundancy in the leakage they offer?
>     - Thank you for your question. In Appendix B, we concatenate the final-layer RoBERTa representations and logits for the output tokens and TP into a single feature vector $Z=[z_{\text{tok}};z_{\text{logit}};z_{\text{TP}}]$ which is then passed to the classifier that estimates mutual information. Figure 1 reports different lower bounds for the four settings (Tok + TP + logit, Tok + logit, Tok + TP, and Tok alone), indicating that logits and TP contribute synergistically to information leakage.
> - Beyond lowering temperature or hiding logits, do you recommend any defense techniques (e.g., randomized decoding, response perturbation) based on your findings?
>     - Thank you for the suggestion. To raise the attack cost by reducing the mutual information $I$, the most straightforward way is to deliberately lower the determinacy of what the attacker can observe; thus, injecting randomness into every stage, such as model outputs, guardrail processing, and the choice of models in an ensemble, is effective.
> - Could your framework also inform differential privacy-style bounds on allowable leakage for LLMs trained with user-sensitive data?
>     - We didn’t quite understand the purpose of this question, so could you please explain it in a bit more detail?
> - Do your bounds generalize to multi-turn conversations where leakage may accumulate or be reinforced across rounds?
>     - As noted in another response, we have proved that our theory holds even without assuming i.i.d. between attacks, so it should remain valid in multi-turn settings. We state in the conclusion that extending the experiments to this is future work.
> - Could your mutual information formulation be extended to LLMs with vision (e.g., GPT-4V)? What are the additional challenges?
>     - Because $Z$ is defined as any attacker-observable signal, such as output tokens, chain-of-thought, image embeddings. Therefore, our theorems extend to multimodal LLMs by simply replacing it with the concatenated vector $Z=[z_{\text{text}};z_{\text{image}}]$. The only additional requirement is to estimate $I(Z;T)$ with a multimodal MI estimator. We will note this extension as future work in the camera-ready version.

---

> > ### Comment · Reviewer_iH4f · 2025-08-05
> >
> > Thank you for your detailed response, which addressed most of my concerns. This score reflects my opinion of the paper.

---

### Note · Authors · 2025-08-16

We sincerely thank the reviewers for their careful evaluation of our work.

We are grateful for the recognition of the paper’s main contributions:
- Theoretical proof of the attack–leakage relationship: We formally define the leakage of “attack-useful information” in model outputs (tokens/reasoning traces/logits) as bits per query, and provide a clear theoretical proof that greater leakage corresponds to a smaller theoretical lower bound on the number of queries required for a successful attack.
- Practical significance of the proof: Our results enable the quantification, in terms of leakage bits, of how factors such as revealing reasoning processes, exposing token probabilities, adjusting generation diversity, and applying rate limits affect the number of queries required for a successful attack. This allows defenders to optimize the transparency–security trade-off based on numerical evidence, and allows attackers to compare the efficiency of their methods against the theoretical limit.

We have also addressed the following points, which will be incorporated into the final version. These refinements do not alter the paper’s claims; rather, they provide further support and clarity:
- Removal of the i.i.d. assumption in the proof: In response to concerns regarding this assumption in the original version, we have added a proof showing that our method remains valid without assuming i.i.d.
- Improved visualizations: We have updated the figures by correcting x-axis labels and adjusting line colors for improved clarity.

---

### Decision · Program_Chairs · 2025-09-17

**Decision:**

Accept (spotlight)

**Comment:**

This paper presents a principled information-theoretic framework for quantifying the efficiency of adversarial attacks on LLMs by bounding query complexity in terms of bits leaked per query, with empirical validation across seven models and three attack types. Its strengths lie in the elegant theoretical contribution (extended beyond i.i.d. to adaptive and multi-turn settings), intuitive unifying metric, broad empirical support, and practical relevance for transparency and safety design. Weaknesses include limited actionable defensive guidance, reliance on lower-bound MI estimators, and modest evaluation on real-world API settings. During the rebuttal, the authors convincingly resolved major concerns—providing independence-free proofs, clarifying that verification and generation are equivalent, correcting mislabeled results, and showing robustness across attack methods—leading all reviewers to update or confirm acceptance. Overall, this is a solid and reproducible contribution of clear significance.